 **eLIFE**

# Auditory synapses to song premotor neurons are gated off during vocalization in zebra finches

**Kosuke Hamaguchi[1†], Katherine A Tschida[1†], Inho Yoon[2], Bruce R Donald[2,3,4], Richard Mooney[1]\***

[1]Department of Neurobiology, Duke University Medical Center, Durham, United States; [2]Department of Electrical and Computer Engineering, Duke University, Durham, United States; [3]Department of Computer Science, Duke University, Durham, United States; [4]Department of Biochemistry, Duke University Medical Center, Durham, United States

**Abstract** Songbirds use auditory feedback to learn and maintain their songs, but how feedback interacts with vocal motor circuitry remains unclear. A potential site for this interaction is the song premotor nucleus HVC, which receives auditory input and contains neurons (HVC$_X$ cells) that innervate an anterior forebrain pathway (AFP) important to feedback-dependent vocal plasticity. Although the singing-related output of HVC$_X$ cells is unaltered by distorted auditory feedback (DAF), deafening gradually weakens synapses on HVC$_X$ cells, raising the possibility that they integrate feedback only at subthreshold levels during singing. Using intracellular recordings in singing zebra finches, we found that DAF failed to perturb singing-related synaptic activity of HVC$_X$ cells, although many of these cells responded to auditory stimuli in non-singing states. Moreover, in vivo multiphoton imaging revealed that deafening-induced changes to HVC$_X$ synapses require intact AFP output. These findings support a model in which the AFP accesses feedback independent of HVC.

**\*For correspondence:** mooney@
neuro.duke.edu

†These authors contributed
equally to this work

**Competing interests:** The
authors declare that no
competing interests exist.

**Reviewing editor:** Ronald L
Calabrese, Emory University,
United States

## Introduction

Many elaborate behaviors, ranging from playing a guitar to engaging in a conversation, depend on the brain's ability to integrate performance-related auditory feedback with the output of motor circuits controlling behavior. The synaptic mechanisms underlying the integration of sensory and motor-related signals remain enigmatic. Songbirds use auditory feedback to learn and maintain their vocalizations (*Konishi, 1965*; *Price, 1979*) and possess well-delineated neural circuits for singing (*Nottebohm et al., 1982*), thus providing an attractive organism in which to identify synaptic mechanisms for auditory–vocal integration. Nevertheless, despite important progress in identifying synaptic mechanisms of auditory-guided song plasticity (*Mooney, 1992*; *Olveczky et al., 2005*; *Andalman and Fee, 2009*; *Roberts et al., 2010*; *Warren et al., 2011*), how auditory feedback is integrated in the brain to affect the neural circuits for singing remains poorly understood.

One site where feedback could interact with song motor commands is the sensorimotor nucleus HVC, which is necessary for singing (*Nottebohm et al., 1976*), exhibits neural activity precisely time-locked to song (*Hahnloser et al., 2002*; *Kozhevnikov and Fee, 2007*; *Prather et al., 2008*), and receives input from auditory regions that contain auditory feedback-sensitive neurons (*Keller and Hahnloser, 2009*). Within HVC, a distinct type of projection neuron (HVC$_X$) provides input to an anterior forebrain pathway (AFP) that is necessary for song learning and that is anatomically similar to cortico-basal ganglia pathways in mammals (*Farries and Perkel, 2002*; *Doupe et al., 2005*). Various studies have shown that the output nucleus of the AFP (the lateral magnocellular nucleus of the anterior

**eLife digest** Whenever we speak, sing, or play a musical instrument, we use auditory feedback to fine-tune our movements to achieve the sound that we want. This same process is used by songbirds to learn and maintain their songs. As juvenile birds practice singing, they compare their vocalizations with their father's song, which they will previously have stored in memory, and continually tweak their own song until the two versions match.

It has been suggested that auditory feedback is integrated with song motor commands—the instructions from the brain to move the muscles required for singing—in a region of the songbird brain called the song premotor nucleus HVC. The structure of certain neurons in this region, known as $HVC_X$ cells, rapidly changes when a bird is deafened, which suggests that these $HVC_X$ cells detect auditory feedback.

Hamaguchi et al. have now tested this idea by using fine electrodes to record the signals in $HVC_X$ cells in male zebra finches as they sang. The cells changed their activity patterns whenever the birds changed their vocalizations. By contrast, these patterns did not change when the birds heard a distorted version of their own song played back to them as they sang. This suggests that $HVC_X$ cells are insensitive to auditory feedback, and that they mainly encode song motor commands instead.

If $HVC_X$ cells don't detect feedback, then why does deafening affect them? $HVC_X$ cells send signals indirectly to a brain region called the LMAN (which is short for the lateral magnocellular nucleus of the anterior nidopallium). Normally, if a bird becomes deaf, the quality of their song begins to deteriorate, but this deterioration can be prevented by destroying the LMAN. Hamaguchi et al. used high resolution imaging to show that destroying the LMAN also prevents deafening from altering the structure of $HVC_X$ cells. Again, this suggests that auditory feedback is not relayed from the HVC to the LMAN; instead the flow of information is in the opposite direction.

This surprising finding—namely, that $HVC_X$ cells do not integrate auditory feedback and song motor commands—raises the question of which brain region is in fact responsible for this process. Further experiments will be required to identify the underlying circuitry in the brains of songbirds.

nidopallium, or LMAN) is the source of behavioral variations that can adaptively bias song in response to feedback perturbations (*Andalman and Fee, 2009*; *Warren et al., 2011*; *Charlesworth et al., 2012*). Specifically, LMAN lesions prevent deafening-induced song degradation (*Brainard and Doupe, 2000*; *Andalman and Fee, 2009*; *Warren et al., 2011*), and inactivating LMAN can reverse shifts in syllable pitch induced by distorted auditory feedback (DAF) (*Brainard and Doupe, 2000*; *Andalman and Fee, 2009*; *Warren et al., 2011*). These findings suggest that the AFP has access to feedback-related information about song performance. One potential source of this information is the $HVC_X$ cell population, because auditory responses that can be detected in the AFP of anesthetized birds disappear when HVC is pharmacologically inactivated (*Roy and Mooney, 2009*). Moreover, individual $HVC_X$ cells can fire in remarkably similar patterns during singing and in response to song playback (*Prather et al., 2008*; *Fujimoto et al., 2011*). However, the singing-related action potential activity of $HVC_X$ cells does not change when auditory feedback is masked by noise bursts (*Kozhevnikov and Fee, 2007*; *Prather et al., 2008*). These and other observations (*Hessler and Doupe, 1999*) are consistent with the idea that in singing birds, $HVC_X$ cells provide the AFP with a motor-related efference copy rather than an auditory feedback signal.

Although extracellular recordings indicate that $HVC_X$ cell output in singing birds is insensitive to feedback perturbations over tens of minutes, a recent imaging study found that deafening shrinks and destabilizes dendritic spines on $HVC_X$ neurons within ~12–48 hr, and these changes precede and predict the severity of song degradation (*Tschida and Mooney, 2012*). Therefore, a remaining possibility is that $HVC_X$ cells receive subthreshold feedback signals during singing that modify their dendritic spines and more slowly alter $HVC_X$ action potential output, and thus AFP activity, over a period of hours to days. One prediction of this model is that synaptic inputs onto $HVC_X$ cells will be acutely sensitive to feedback perturbation. In this study, we tested this idea using sharp intracellular current clamp recordings in freely singing zebra finches exposed to DAF. A second prediction of this model is that the deafening-induced changes in $HVC_X$ spines will be caused by feedforward changes to HVC's auditory afferents rather than by AFP-dependent mechanisms that drive deafening-induced vocal plasticity. To test this

prediction, we imaged HVC$_X$ dendritic spines in zebra finches that received LMAN lesions prior to deafening.

## Results

### Singing-related subthreshold activity of HVC$_X$ cells is insensitive to DAF

To determine whether synaptic inputs onto HVC$_X$ cells convey auditory feedback signals during singing, we made intracellular sharp electrode recordings from HVC neurons in unrestrained, young adult (~95 days post hatch [dph]) male zebra finches using a modified version of recording techniques developed recently (*Lee et al., 2006*; *Long et al., 2010*) (*Figure 1—figure supplement 1*). We recorded from a total of 72 HVC$_X$ neurons in 11 birds as they engaged in spontaneous bouts of singing produced in social isolation (i.e., undirected song) and/or listened to playback of the bird's own song (BOS). HVC$_X$ cells were identified either by antidromic stimulation methods (*Figure 1A,B*), DC current-evoked firing patterns (*Mooney, 2000*), or characteristic singing-related hyperpolarization (*Long et al., 2010*). As previously reported (*Long et al., 2010*), all HVC$_X$ cells we recorded without current injection showed spontaneous, regular action potential activity when the bird was not vocalizing (8.2 ± 5.8 Hz, mean ±SD), entered a hyperpolarized state within several hundred milliseconds before song onset (baseline −58.5 ± 7.1 mV, during singing −62.3 ± 7.3 mV), and all except two exhibited one or more action potential bursts during the utterance of the stereotyped sequence of syllables constituting the song motif (*Figure 1D*, *Video 1*). Each cell's pattern of subthreshold membrane potential activity (V$_m$) and action potential bursts were highly stereotyped from one motif to the next and from bout to bout (*Figure 1D*).

Although previous studies have shown that singing-related action potentials of HVC$_X$ neurons are insensitive to DAF (*Kozhevnikov and Fee, 2007*; *Prather et al., 2008*), we tested whether singing-related synaptic inputs to HVC$_X$ cells encode auditory information by perturbing auditory feedback during singing. To distort the bird's experience of singing-related auditory feedback, we used a computer-controlled real-time system to detect a specific syllable in the bird's song motif and trigger sound playback during the production of the ensuing 'target' syllable (*Figure 1C*; 'Materials and methods' and *Figure 1—figure supplement 2*). Playback sounds included either a 100-ms noise burst or a recorded version of one of the bird's own syllables; using either of these stimuli to distort singing-related auditory feedback has been shown to induce gradual changes to adult song, indicating the existence of neural circuitry that detects these acute feedback perturbations and induces vocal plasticity (*Leonardo and Konishi, 1999*; *Tumer and Brainard, 2007*; *Andalman and Fee, 2009*). To establish the efficacy of this DAF method, we set the sound amplitude slightly above the level that causes song truncation in some initial trials (~65 dB at the center of the cage). To ensure that any changes in subthreshold activity we might detect were driven by feedback perturbations rather than by acute changes in motor-related activity, we excluded from analysis any trials in which DAF triggered truncation or changes in song tempo (see below for a description of subthreshold activity in those DAF trials that evoked acute motor effects; also see *Sakata and Brainard, 2008*). In separate behavioral experiments, we also confirmed that applying our perturbation protocol in a pitch-contingent manner over a period of days as previously described (*Tumer and Brainard, 2007*) was sufficient to shift the pitch of target syllables (*Figure 1—figure supplement 3*; the observed hit rate was 30–70%).

To measure any rapid effects of DAF on the singing-related synaptic activity of HVC$_X$ cells, we compared membrane potential records from randomly interleaved trials with ('hit') and without ('catch') DAF (*Figures 1D and 2A*; target hit rate = 50%; observed rate was 30–70%). In a subset of cells, we used current injection to strongly hyperpolarize the membrane potential to maximize our ability to detect feedback-dependent changes in inhibitory inputs. To quantify the amplitude and timing of singing-related synaptic activity,

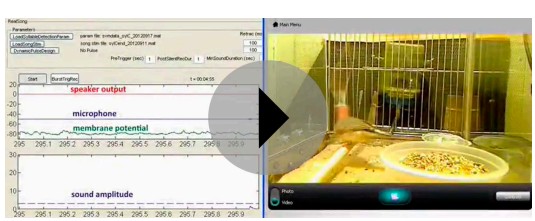

**Video 1.** Sharp intracellular current clamp recording made in an identified HVC$_\mathbf{x}$ neuron from a freely behaving and singing male zebra finch. Left; microphone (blue), membrane potential (green), and speaker output (red) updated at 1 Hz. Right; simultaneously monitored image recorded through a webcam in the recording chamber. In this movie, sound is recorded through the webcam. The whole movie is played in normal speed (x1). Injection current was changed from negative (−0.67 nA), to zero current (0 nA).

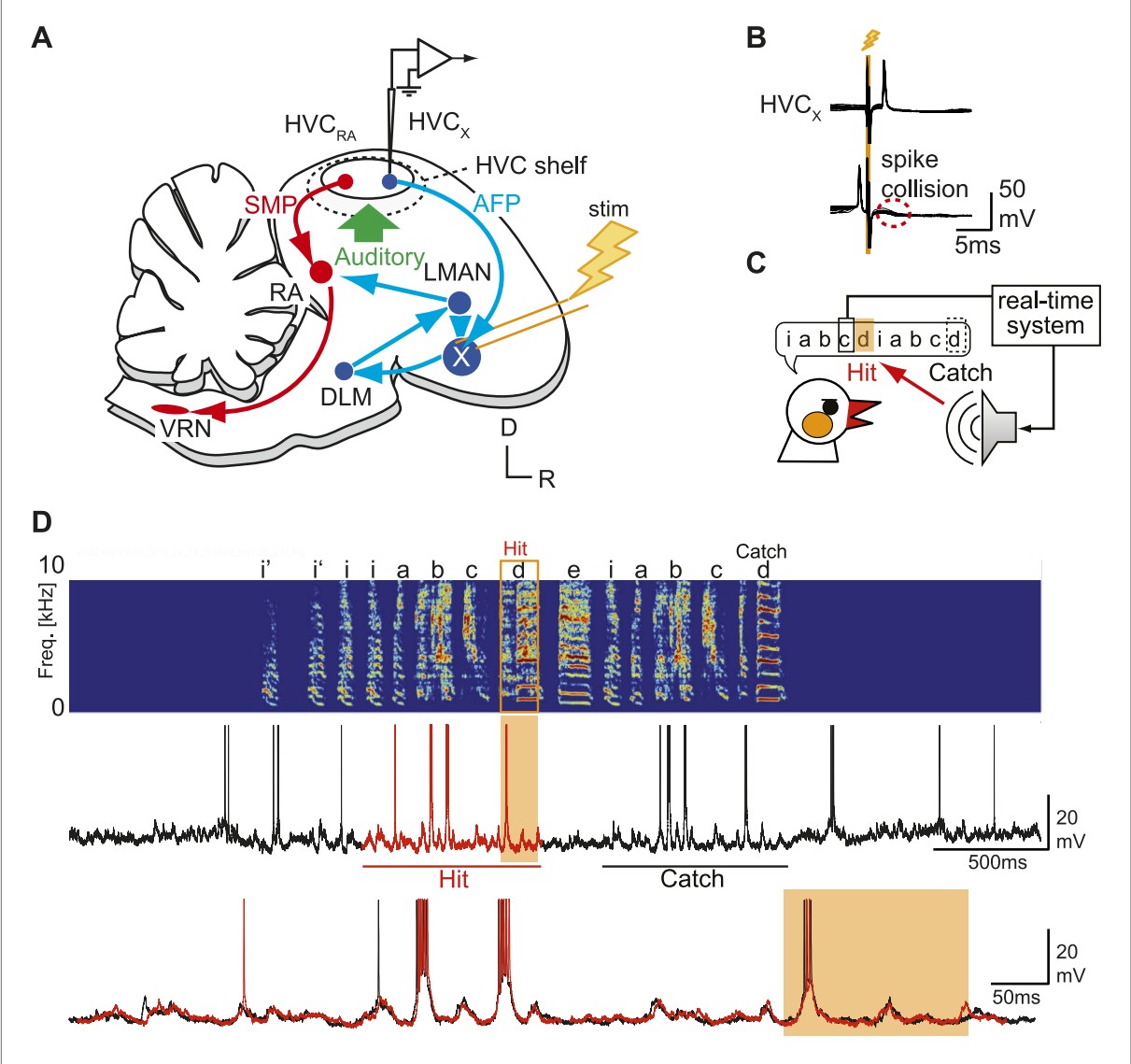

**Figure 1**. Sharp intracellular recordings from sensorimotor neurons in singing birds. (**A**) Schematic illustrates the configuration of the in vivo intracellular recording methods used to measure subthreshold activity of $HVC_X$ neurons. Abbreviations: AFP, anterior forebrain pathway (light blue); SMP, song motor pathway (red); RA, robust nucleus of arcopallium; DLM, dorsolateral division of the medial thalamus; LMAN, lateral magno-cellular nucleus of the anterior nidopallium; VRN, brain stem vocal respiratory network, which includes the tracheosyringeal portion of the hypoglossal motor nucleus (nXIIts) and the respiratory premotor neurons located in the rostral ventrolateral medulla (RVL); R, rostral; D, dorsal. (**B**) Antidromic identification of $HVC_X$ neurons was achieved by electrically stimulating Area X combined with spike collision tests. (**C**) The experimental design used to generate distorted auditory feedback (DAF). Shortly (~8 to 10 ms) after detecting that the bird sang the target syllable, a recorded version of the target syllable or a noise burst was played to the bird through a speaker (hit); DAF was suppressed on randomly chosen trials (catch). (**D**) Examples of $HVC_X$ intracellular membrane potential recordings during hit and catch conditions. Top: sonogram. Middle: simultaneously recorded $HVC_X$ neuron membrane potential. Bottom: expanded view of membrane potential traces aligned to the onset of the entire motif (iabcd), which was sung twice in this bout. The timing of DAF is shown in boxed and shaded regions.

The following figure supplements are available for figure 1:

**Figure supplement 1**. The design of the integrated intracellular microdrive used in these experiments.

**Figure supplement 2**. Flow of pre-target and target syllable detection.

**Figure supplement 3**. Contingent DAF drives adaptive changes in spectral features of the target syllable.

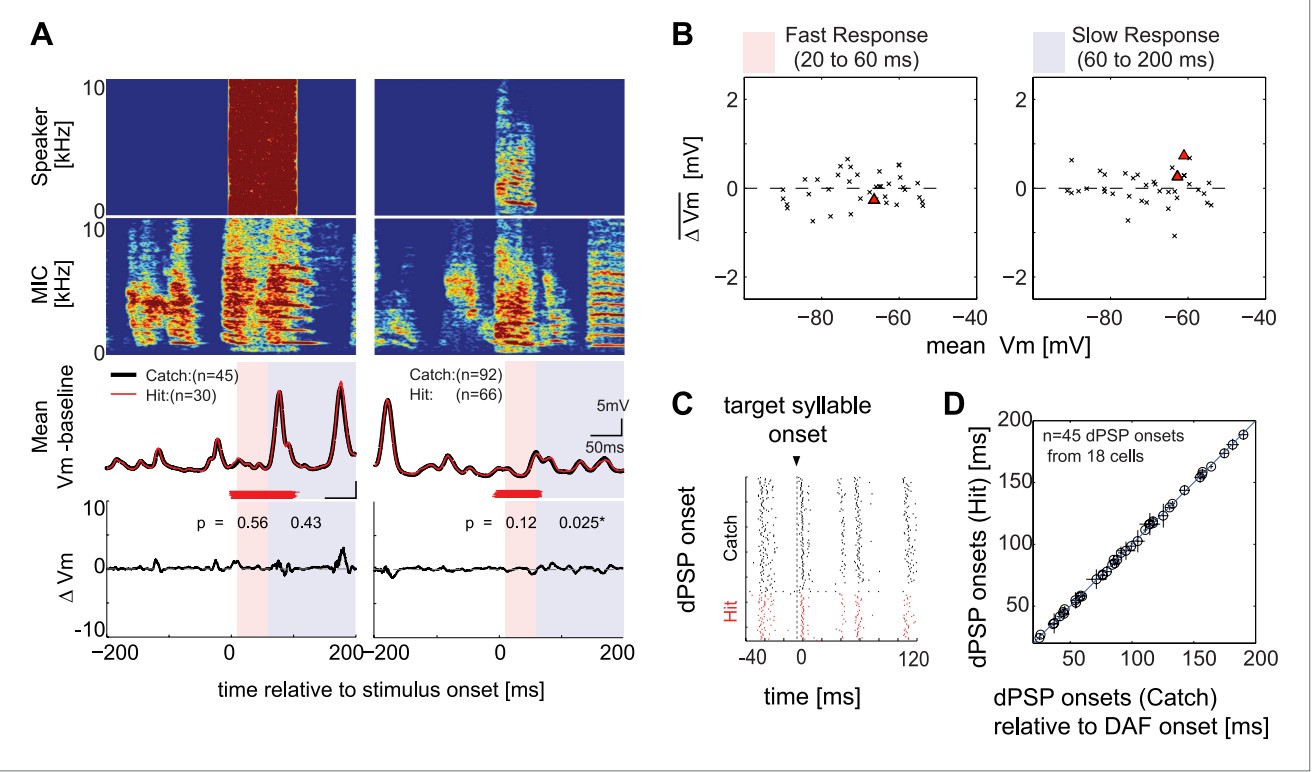

**Figure 2**. Synapses onto HVC$_X$ neurons do not convey auditory feedback signals during singing. (**A**) Each row represents (from top to bottom) the speaker output (Speaker), microphone input (MIC), trial-averaged membrane potential activity in hit (red) and catch (black) trials relative to the baseline (defined as −200 to 0 ms before DAF onset), and the difference in membrane potential between hit and catch conditions (ΔV$_m$). Two examples of HVC$_X$ singing-related subthreshold activity are shown. Two time ranges are set to bracket DAF onset and offset (fast: 20–60 ms, red-shaded region covers the fastest synaptic latency of HVC in response to auditory stimuli [*Lei and Mooney, 2010*]; slow: 60–200 ms, purple shaded region covers the remainder of the DAF period). Red horizontal lines indicate the timing of the DAF stimulus presented during each electrophysiological recording. Data are aligned to the onset of the target syllable, and time zero is set to the mean DAF onset. (**B**) Population analysis of time-averaged ΔV$_m$ in fast and slow response windows showed no significant changes in response to DAF (n = 38 comparisons made from n = 34 cells; four cells were analyzed at both resting and at hyperpolarized membrane potentials. Individual cell-based analysis revealed that all except two cells showed non-significant changes; triangles [p<0.05], crosses [p≥0.05, paired *t*-test]). (**C**) Examples of dPSP onset timings aligned to target syllable onset obtained from a single cell. (**D**) Onset timings of dPSPs measured in catch vs hit trials were indistinguishable (n = 45 dPSPs [1–4 dPSP onsets per cell] from 18 cells; see 'Materials and methods' for more information about dPSP onset detection and clustering methods used for peak detection; mean p=0.52, min p=0.06, *t*-test).

we computed the average deviation of the baseline subtracted membrane potential between hit and catch trials (ΔV$_m$; see 'Materials and methods') and estimated the onset times of depolarizing post-synaptic potentials (dPSPs) 20–200 ms after DAF onset, which brackets the auditory-evoked synaptic latencies of HVC neurons (*Lei and Mooney, 2010*) and the offset of the DAF stimulus (*Figure 2A–D*). As a population, HVC$_X$ cells exhibited no systematic changes in ΔV$_m$ or dPSP onset times in the presence of DAF (*Figure 2B–D*; n = 34 cells). When treated on a cell-by-cell basis, none showed changes in dPSP onset times, and only two cells showed slight (<1 mV) but significant changes in ΔV$_m$ (0.01 ≤ p<0.05; see *Figure 2A*, *right*, for one of these positive cases), an outcome that could be accounted for by a false positive rate for this sample size (38 comparisons; 4 cells were sampled at two different 'resting' potentials). Taken together, these results indicate that fast synaptic inputs onto HVC$_X$ neurons are insensitive to auditory feedback perturbation during singing.

## DAF-insensitive HVC$_X$ cells can respond to auditory stimuli in non-singing states

Although previous recordings made in anesthetized zebra finches indicate that most HVC$_X$ cells can respond to auditory stimulation, and the sample of HVC$_X$ cells that we tested with DAF is reasonably large (n = 34), it remains possible that we sampled entirely from a subset of cells that do not receive

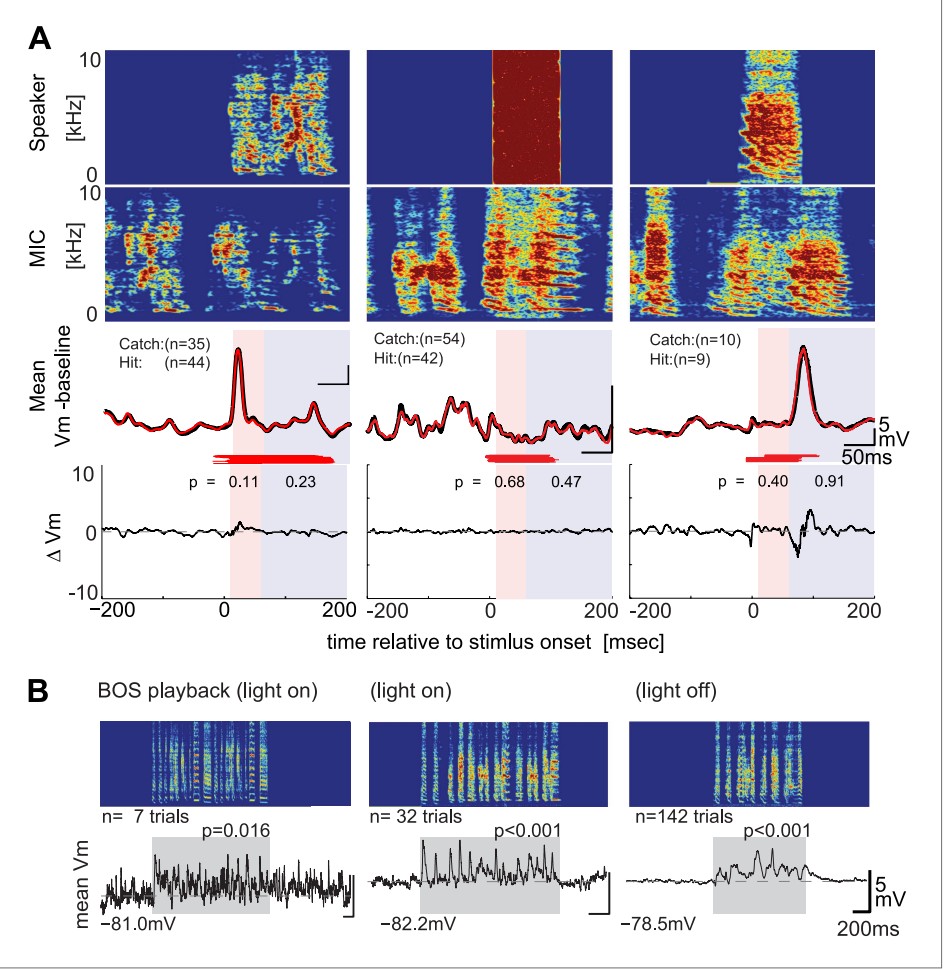

**Figure 3**. DAF-insensitive HVC_X cells can respond to auditory stimuli in non-singing states. (**A**) Examples of trial-averaged singing-related subthreshold activity of DAF-insensitive cells in hit and catch conditions, following the same scheme shown in **Figure 2A**. (**B**) Examples of averaged subthreshold responses to BOS playback in the cells shown in (**A**) in day (light on) or night (light off) conditions. The significance of the BOS-evoked auditory response measured in non-singing states was determined either by the difference in the mean (Mann–Whitney $U$-test) or standard deviations (Ansari–Bradley test, a non-parametric test of variance) of the membrane potential fluctuations during the stimulus period and a pre-stimulus baseline period.

The following figure supplements are available for figure 3:

**Figure supplement 1**. Summary of auditory responses of HVC_X cells in non-singing states.

auditory input. However, we found that DAF-insensitive HVC_X cells could respond to auditory presentation of the bird's own song during periods when the bird was not singing, including during the daytime or minutes to hours into the night (**Figure 3A–B**; 7/8 DAF-insensitive cells showed significant subthreshold responses to BOS playback during the day, and 4/4 DAF-insensitive cells that we tracked across the day–night boundary showed significant sub- and suprathreshold responses during darkness). Moreover, the majority of HVC_X cells we recorded from, including those in which we did not collect singing-related activity, displayed BOS-evoked auditory activity during the day or night (**Figure 3—figure supplement 1**, subthreshold responses: n = 15/27 cells in day; 10/11 cells in night; suprathreshold responses: 5/12 cells in day; 7/8 cells in night). Indeed, the increased proportion of auditory-responsive HVC_X cells we detected during the night is consistent with prior extracellular studies that show auditory responses in the zebra finch HVC are strongly state-dependent (**Cardin and Schmidt, 2003**; **Rauske et al., 2003**; **Nick and Konishi, 2005**). Therefore, we presume that many of the DAF-insensitive cells that we recorded receive auditory input but that these inputs are not activated during singing.

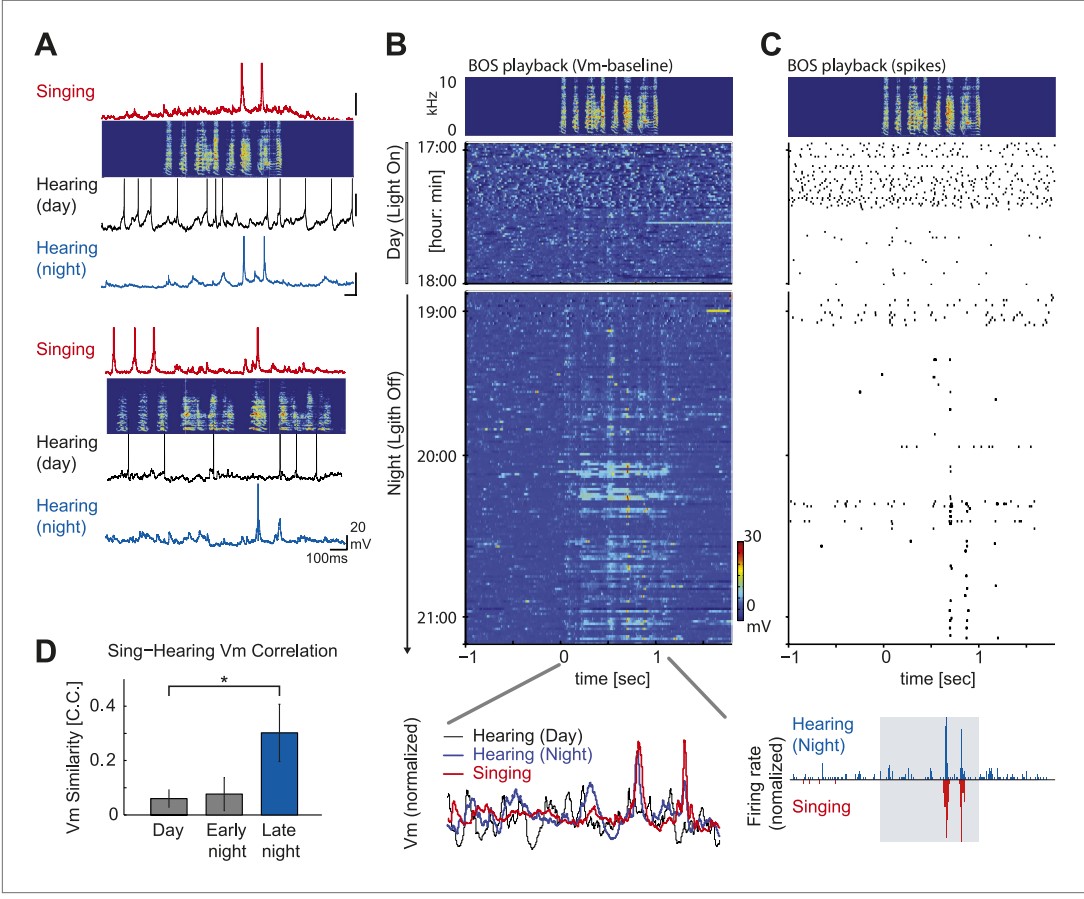

**Figure 4**. Auditory-vocal mirror neurons in zebra finches. (**A**) Examples of singing and auditory-related membrane potential activity of two HVC_X cells aligned to each bird's motif. Initially weak but significant auditory responses during the day (black) become robust during the night (blue) and precisely mirror the cell's singing-related activity (red). The cell at the top is the same cell shown in **Figure 3** (**A**) right, which lacked DAF sensitivity. (**B**) Emergence of robust BOS-evoked subthreshold activity as the night progresses. Top: BOS responses near the day–night boundary (same cell shown in **Figure 4A**, top). Baseline-subtracted membrane potential responses aligned to the BOS playback (top panel) and shown in the order of the recording time. Bottom: normalized voltage trace of the cell's BOS evoked activity at day (black), night (blue) and singing-related activity (red). (**C**) The same cell's action potential response to BOS playback, plotted in the same scheme used in (**B**). 1–2 hr after nightfall, strong BOS-evoked action potential activity was detected that closely mirrored the singing-related activity recorded from this same cell 4–5 hr earlier, during the daytime. (**D**) Similarity of a cell's subthreshold activity during singing and hearing the same motif in the day or night measured by the correlation coefficient (C.C.) of averaged membrane potential records reveals enhanced similarity of singing-related and BOS-evoked activity later in the night (≥30 min after light off). Data are only from auditory-responsive cells in which we also collected singing data (singing vs hearing data during the day: n = 7, C.C. = 0.076 ± 0.041 (SEM); Early night, < 30 min after light off: n = 7, C.C. = 0.077 ± 0.061; Late night, ≥ 30 min after light off: n = 4, C.C. = 0.302 ± 0.105. Mann–Whitney U-test, p=0.04 compared to day).

Prior studies in other songbirds indicate that a subset of HVC_X cells display a strong correspondence in their auditory and motor-related activity (**Prather et al., 2008**; **Fujimoto et al., 2011**), and this sensorimotor 'mirroring' is speculated to arise from the integration of feedback- and motor-related signals by these cells (**Hanuschkin et al., 2013**). Although prior studies have not described auditory–vocal mirroring in HVC of zebra finches, longer intracellular recordings (>3 hr) that we made in two DAF-insensitive HVC_X cells revealed clear evidence of auditory–vocal mirroring (**Figure 4A,B**). Specifically, at later times during the night, we detected BOS-evoked sub- and supra-threshold activity that closely resembled the singing-related activity recorded earlier during the daytime. Moreover, in a larger subset of HVC_X cells in which we recorded singing-related activity with or without DAF during the day and tracked across the day–night boundary, we found that the correlation in membrane potential

activity recorded during singing and in response to BOS playback increased at later times in the night (*Figure 4C*). These findings show that auditory–vocal mirror neurons in the zebra finch HVC are insensitive to auditory feedback perturbation during singing and also suggest that sensorimotor mirroring may be a latent feature of a large cohort of $HVC_X$ cells in this species.

## Intracellular methods can detect DAF-related synaptic activity in cells ventral to HVC

A remaining issue is whether the intracellular recording methods we used can detect small subthreshold membrane potential fluctuations in cells that are likely to receive input from DAF-sensing cells. In a

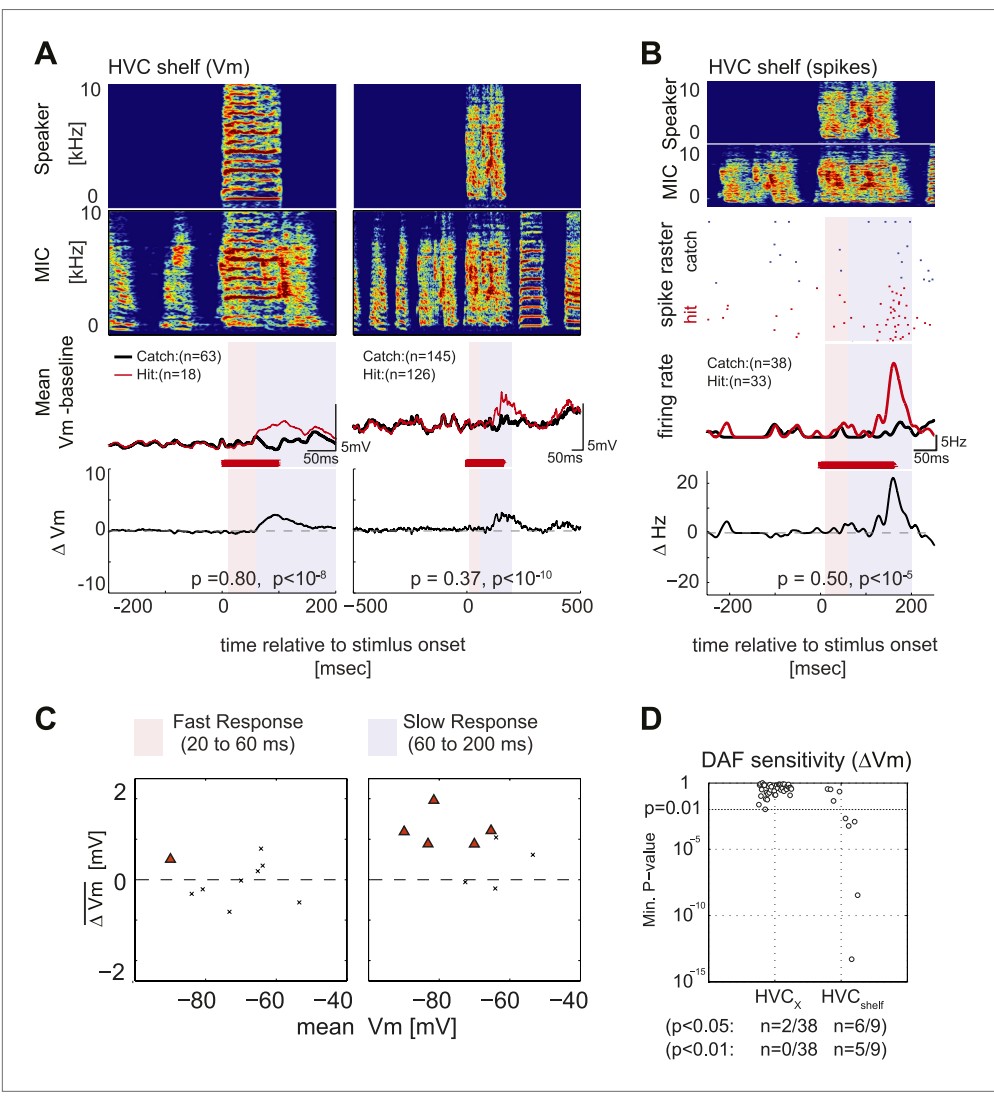

**Figure 5**. Synapses onto HVC shelf neurons convey DAF signals during singing. (**A**) Examples of trial-averaged subthreshold activity patterns of DAF-sensitive HVC shelf cells in hit and catch conditions, following the same scheme as *Figure 2A*. Exposure to DAF significantly increased subthreshold depolarizations during singing. (**B**) An HVC shelf cell that showed a significant suprathreshold response to DAF, following the same scheme shown in *Figure 2A* except for the addition of a spike-rastergram in the second row. (**C**) Time-averaged $\Delta V_m$ in fast and slow response windows (n = 9 cells, triangles [p<0.05], crosses [p≥0.05, paired t-test]). (**D**) DAF sensitivity plot for HVC shelf neurons and $HVC_X$ cells. Nearly half of the HVC shelf neurons showed DAF sensitivity. The minimum p-values of time-averaged $\Delta V_m$ either in fast or slow time window are plotted, Mann–Whitney *U*-test.

The following figure supplements are available for figure 5:

**Figure supplement 1**. HVC shelf cells show subthreshold DAF sensitivity.

small subset of experiments, we made intracellular recordings (n = 14 neurons in 5 birds) in the region immediately ventral to HVC (i.e., HVC shelf, *Figure 5—figure supplement 1A*), which is known to receive input from auditory regions (*Fortune and Margoliash, 1995*; *Vates et al., 1996*) that contain DAF-sensitive neurons (*Keller and Hahnloser, 2009*). During singing, we found that most (5/9) HVC shelf cells showed significantly greater subthreshold depolarizing responses in the presence of DAF, consistent with the notion that they receive synaptic input from neurons that exhibit DAF-sensitive action potential output (*Figure 5A,C, D*; one of these cells also showed a significant suprathreshold response to DAF, *Figure 5B*). These recordings also revealed that synaptic activity in HVC shelf cells can be modulated during singing, although the pattern of this activity was less stereotyped from one motif to the next than observed in HVC$_X$ cells (*Figure 5—figure supplement 1B*. HVC shelf: n = 14 cells, C.C. = 0.12 ± 0.03 (SEM). HVC$_X$ cells: n = 53 cells, C.C. = 0.70 ± 0.02, without time warping). These data confirm that the intracellular recording technique used here can detect subthreshold membrane potential responses to DAF in the singing zebra finch, further strengthening the idea that auditory-related synaptic inputs to HVC$_X$ cells are gated in an 'off' position during singing.

## Evidence that HVC$_X$ cells encode motor-related information

Our findings indicate that HVC$_X$ cells are insensitive to auditory feedback perturbation during singing, raising the possibility that they primarily encode motor-related information. One expectation of a motor-related signal is that it should precede sound onset during singing. To test this idea, we examined the relationship between the onsets of dPSPs and syllables during bouts of singing (*Figure 6A–C*). Population-averaged cross-correlation analysis revealed that dPSP onsets in HVC$_X$ cells occur on average ~25 ms before syllable onsets (n = 53 cells, *Figure 6D*), similar to the findings from prior extracellular studies (*Kozhevnikov and Fee, 2007*; *Fujimoto et al., 2011*). In contrast, dPSP onsets were not correlated with syllable offsets (*Figure 6D*), indicating that singing-related synaptic inputs to HVC$_X$ cells are correlated to future, rather than ongoing or past, vocal output. Another expectation of a motor-related signal is that, should change in parallel to changes in vocal output. As previously mentioned, DAF can trigger acute effects, such as truncation or slowing of song tempo (*Figure 6E,G*) (*Sakata and Brainard, 2008*), which are distinct from the slower changes in song that are associated with feedback-dependent error correction mechanisms. In fact, we observed significant changes in membrane potential fluctuations of HVC$_X$ cells when DAF-triggered song truncation or delays in syllable onsets (*Figure 6F,H*). Therefore, the intracellular method we used is capable of detecting DAF-related changes in the sub-threshold activity of HVC$_X$ cells, but only when DAF triggers acute vocal motor effects. Taken together, these findings suggest that synaptic inputs to HVC$_X$ cells do not convey auditory feedback signals during singing and instead encode information about future vocal output, consistent with a motor-related signal.

## Deafening-induced changes to HVC$_X$ spines depends on an intact AFP

Our finding that HVC$_X$ cells lack feedback-sensitive synaptic activity during singing raises the question of how deafening drives structural and functional changes to synapses on HVC$_X$ cells that precede song degradation (*Tschida and Mooney, 2012*). One possibility is that feedback perturbations are detected by auditory neurons indirectly presynaptic to HVC$_X$ cells, and information from these feedback-sensing cells alters synapses on HVC$_X$ cells more gradually as a prelude to the AFP-mediated error correction processes that result in vocal plasticity (*Figure 7A*). In this 'feedforward model', disrupting auditory feedback should be sufficient to induce synaptic changes in HVC$_X$ cells regardless of any downstream processes in the AFP. Another possibility is that deafening-induced changes to synapses on HVC$_X$ cells depend on downstream processes in the AFP that are known to be critical for feedback-dependent changes to song (*Figure 7B*). Indeed, contrary to the long-held assumption that activity only propagates from HVC to the AFP, a recent study established that activity could propagate from LMAN through recurrent circuitry to HVC (*Hamaguchi and Mooney, 2012*). Therefore, a remaining possibility is that deafening-induced changes to synapses on HVC$_X$ cells depend on LMAN activity and not simply on the removal of auditory feedback to HVC, as predicted by the feedforward model.

To distinguish between these possibilities, we used multiphoton in vivo imaging methods to monitor the size and stability of dendritic spines on HVC$_X$ cells in adult male zebra finches that received bilateral LMAN lesions prior to surgical deafening (n = 3 male zebra finches, 120–140 dph). As previously described (*Roberts et al., 2010*; *Tschida and Mooney, 2012*), a GFP-lentivirus was injected into HVC

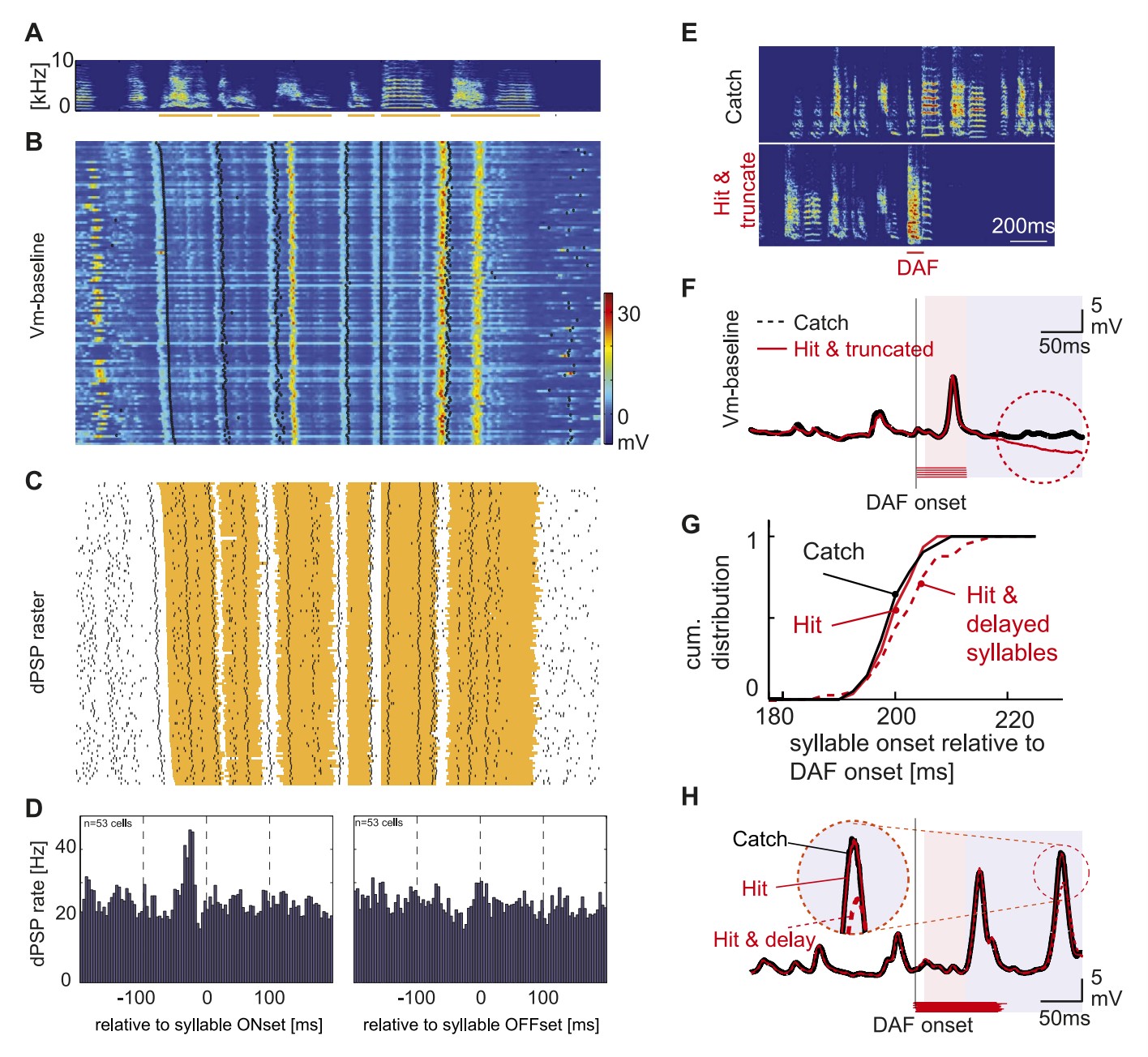

**Figure 6**. Synaptic inputs onto HVC$_X$ cells encode future syllable onsets. (**A**) An example of a stereotyped syllable sequence, with syllables underlined by orange bars. (**B**) Pseudo-color representation of the membrane potential patterns relative to baseline while the bird sang many repetitions of the syllable sequence shown in (**A**). Black dots; syllable onsets. Data are aligned to a specific syllable and sorted from long to short motifs, which reflect natural variations in zebra finch song tempo. (**C**) Same data shown in (**B**), but dPSP onsets are shown by black dots, and the timing of individual syllables is represented by the orange regions. (**D**) Syllable onset- and offset-triggered average of dPSP onset rate (n = 53 HVC$_X$ cells). (**E**) Example spectrograms from catch and 'hit-and-truncated' songs. (**F**) Trial-averaged subthreshold activity during hit-and-truncated songs showed a clear deviation from activity during catch trials. (**G**) Cumulative histogram of syllable onset timings relative to DAF onset in hit trials including delayed syllables (red dashed line), hit trials without delayed syllables (red solid line), and catch trials (black line). (**H**) Examples of trial-averaged subthreshold activity reveal that delayed motor output drives changes in the subthreshold activity (compare black and red dashed lines). After removing hit trials with delayed syllables, the subthreshold activity in hit and catch conditions is nearly identical (black vs red solid lines).

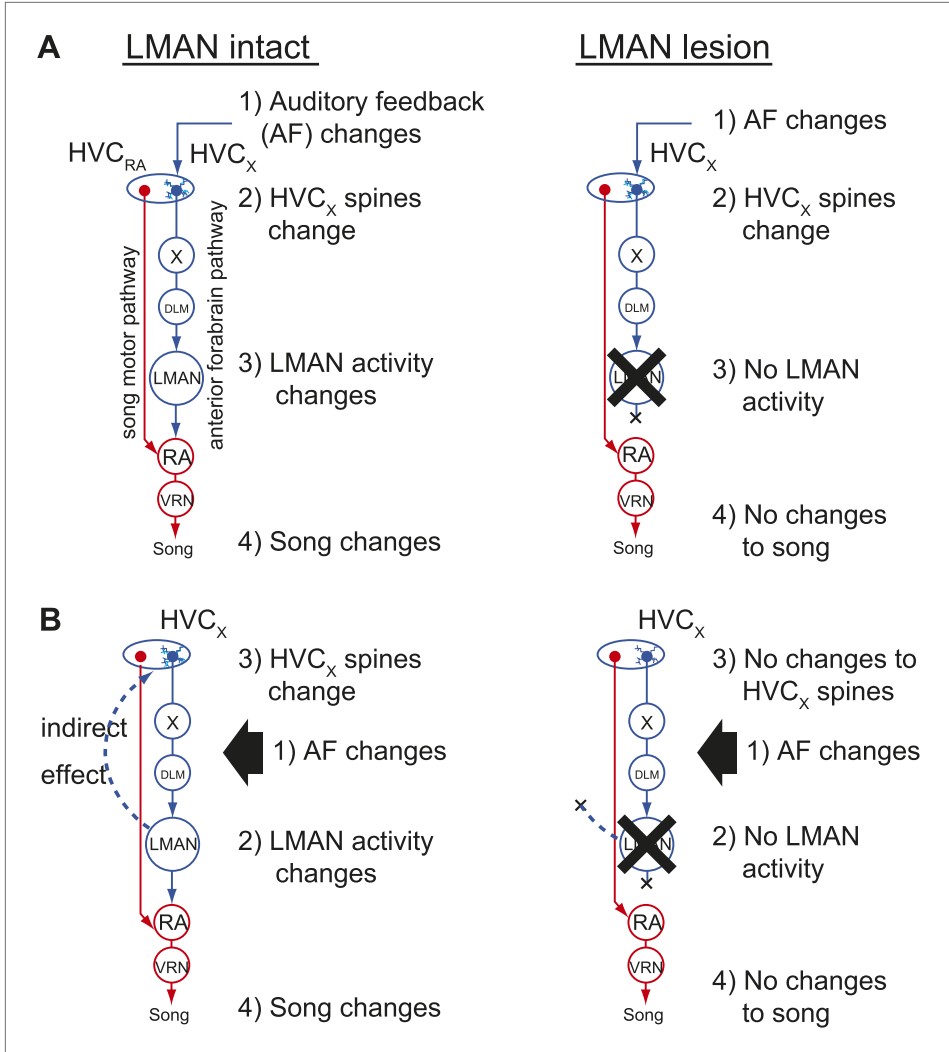

**Figure 7**. Two models of auditory feedback-dependent vocal and HVC dendritic plasticity. (**A**) A model where deafening triggers slow changes to HVC$_X$ spines, subsequently altering AFP activity, which in turn drives vocal plasticity. In this model, LMAN lesions prevent deafening-induced song degradation but will not prevent deafening-induced changes to HVC$_X$ spines. (**B**) A model where deafening acts through LMAN to trigger song plasticity and also to drive changes to HVC$_X$ spines. In this model, LMAN lesions will prevent both deafening-induced song degradation and deafening-induced changes to HVC$_X$ spines.

to label HVC cells, and a retrograde tracer was injected bilaterally into Area X to facilitate identification of HVC$_X$ cells. Additionally, we placed bilateral lesions in LMAN with ibotenic acid 4–5 days before the first imaging session, which was conducted during the bird's subjective nighttime (*Figure 8A*). The morning following the first imaging session, the birds were deafened by bilateral surgical removal of the cochlea, which triggers a form of AFP-dependent song degradation that we confirmed here can be dramatically reduced by prior LMAN lesions (*Figure 8A*; only syllables that underwent significant degradation within the first 3d post-deafening are shown; p values reported in *Figure 8A* for difference in magnitude of spectral change between LMAN lesion and LMAN intact birds; also see below). Surprisingly, in addition to greatly reducing song degradation, LMAN lesions completely prevented the decrease in HVC$_X$ cell spine size index that normally follows deafening (*Figure 8B–C*; 471 spines from 7 HVC$_X$ neurons in 3 LMAN lesion-deafened birds; 495 spines from 7 HVC$_X$ cells in 6 LMAN-intact deafened birds; spine data from LMAN intact-deafened birds were previously reported in *Tschida and Mooney, 2012*; p=0.001 for difference between groups across all time bins). In fact, the post-deafening spine size index values from HVC$_X$ cells in deafened birds with LMAN lesions measured here did not

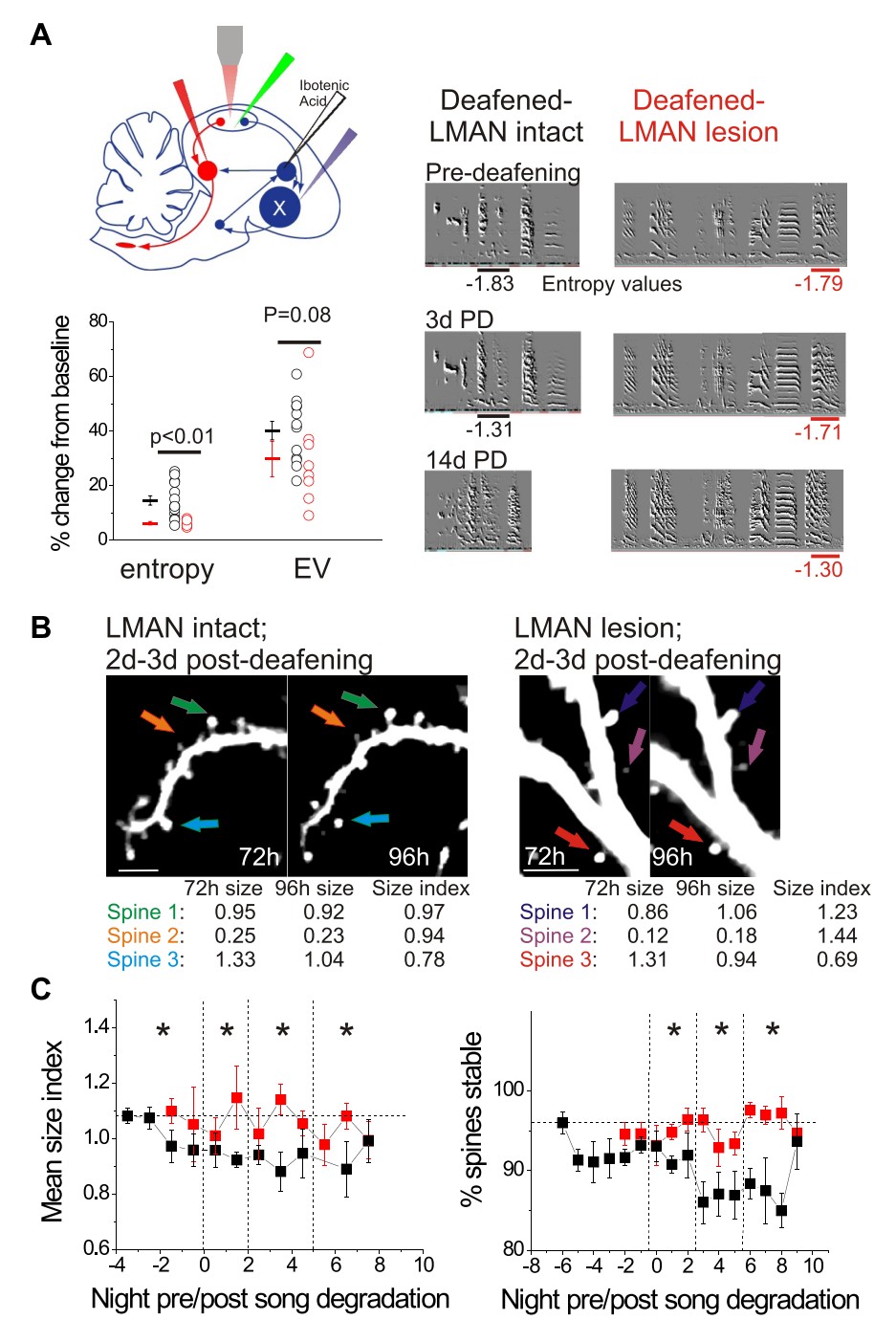

**Figure 8**. LMAN lesions prevent decreases in spine size and spine stability in HVC$_X$ neurons following deafening. (**A**) Upper left: schematic illustrates surgical manipulations for in vivo imaging experiments. Lower left: quantification of deafening-induced changes to syllable entropy and EV at 3 days post-deafening in LMAN intact (black) and LMAN lesion (red) birds (only syllables that underwent significant degradation by 3 days post-deafening are shown; song degradation analyzed in a total of 6 LMAN lesion birds (3 used for imaging) and 19 LMAN intact birds (13 used for imaging in **Tschida and Mooney, 2012**); entropy: 14 syllables from 10 LMAN intact birds, 5 syllables from 5 LMAN lesion birds; EV: 12 syllables from 9 LMAN intact birds, 8 syllables from 4 LMAN intact birds). Right: representative songs before and after deafening from LMAN lesion and LMAN intact birds. Deafened birds that received LMAN lesions still undergo subtle but significant song degradation. (**B**) Representative in vivo, two-photon images of HVC$_X$ neurons showing changes in spine size between 2 and 3 days post-deafening in deafened birds with or

*Figure 8. Continued on next page*

*Figure 8. Continued*

without LMAN lesions (images of HVC$_X$ dendrites from LMAN intact bird are reprinted with permission from *Figure 1B* in *Tschida and Mooney, 2012*, Neuron, copyright Elsevier 2012, All Rights Reserved). LMAN lesions prevent the decrease in the size of HVC$_X$ spines that normally follows deafening (size index <1 in LMAN intact birds). Scale bars, 5 µm. (**C**) Spine size index (left) and spine stability (right) is significantly higher in deafened birds with LMAN lesions. Quantification of spine size (left) and stability (right) in HVC$_X$ neurons from deafened birds with (red) and without (black) LMAN lesions (LMAN intact data were previously reported in *Tschida and Mooney (2012)*, except the values reported here have not been normalized to baseline, pre-deafening measurements). Time bins with significant differences between intact and LMAN lesion groups are indicated with asterisks (p<0.05).

differ from values measured from longitudinally imaged birds with normal hearing (data not shown; spine data from birds with normal hearing were previously reported in *Tschida and Mooney, 2012*). Lesions to LMAN also prevented deafening-induced decreases in HVC$_X$ cell spine stability, and spine stability measurements from HVC$_X$ cells in LMAN lesion-deafened birds were significantly higher than those from LMAN intact-deafened HVC$_X$ cells in all post-song degradation time bins (*Figure 8C*; 2401 spines from 7 HVC$_X$ cells in 3 LMAN lesion-deafened birds; 3562 spines from 14 HVC$_X$ cells in 9 LMAN intact-deafened birds; spine data from LMAN intact-deafened birds were previously reported in *Tschida and Mooney, 2012*; p≤0.02 for difference between groups in last 3 time bins). Therefore, in contrast to a feedforward model of auditory feedback processing by HVC, deafening-induced changes to dendritic spines on HVC$_X$ cells depend on an intact LMAN.

We also conducted several control measurements to exclude the possibility that LMAN lesions by themselves trigger reactive changes in HVC circuitry or to the bird's song that might occlude or otherwise alter spine dynamics in HVC. First, spine dynamics in HVC$_X$ cells following LMAN lesions but prior to deafening did not differ from values obtained from similarly aged birds with normal hearing and an intact LMAN (*Figure 9A*; HVC$_X$ cell spine size index: 1.10 ± 0.06 in LMAN lesion group; 1.07 ± 0.03 in LMAN intact group, mean ±SEM, p=0.62; HVC$_X$ cell spine stability: 94.0 ± 1.1% in LMAN lesion group, 92.0 ± 1.6% in LMAN intact group, p=0.61; LMAN intact data previously reported in *Tschida and Mooney, 2012*). Second, prior to deafening, LMAN lesions did not affect the mean or variability of syllable entropy or entropy variance (EV), two features that are sensitive to small changes in song's acoustic structure (*Figure 9B*; Wilcoxon signed-ranks test used to compare pre-lesion to post-lesion values for 25 syllables from 6 birds [includes 3 birds used for imaging]; data shown only for mean entropy and EV; p=0.09 for mean entropy, p=0.92 for coefficient of variation [CV] of entropy, p=0.25 for mean

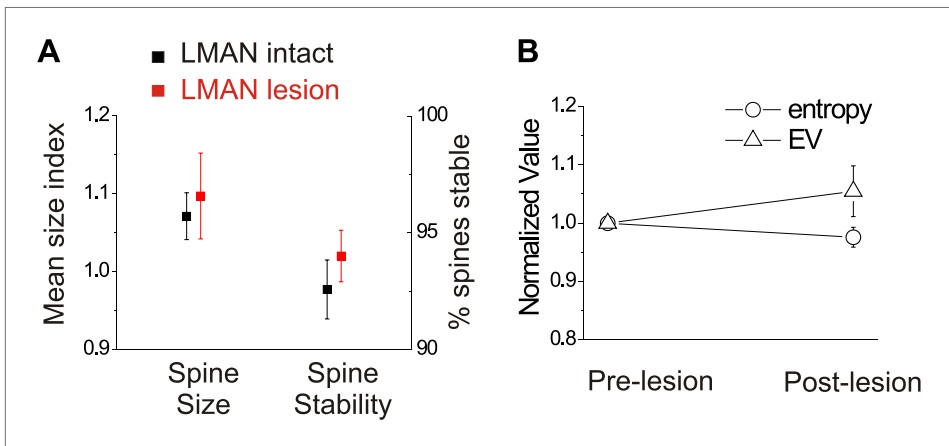

**Figure 9**. In birds with normal hearing, LMAN lesions do not significantly affect HVC$_X$ spine dynamics or spectral features of song. (**A**) LMAN lesions affect neither spine size index nor spine stability in HVC$_X$ neurons (p=0.62 for size, p=0.61 for stability; LMAN intact data were previously reported in *Tschida and Mooney, 2012*). (**B**) LMAN lesions do not affect mean syllable entropy or EV (pre-lesion songs compared to songs from 4 to 5 days post-lesion; 25 syllables from 6 birds, p=0.09 for difference in mean entropy, p=0.25 for mean EV.

EV, p=0.31 for CV of EV). Third, a post mortem analysis of brain tissue indicated that the lesions we made in LMAN were comparable in volume to those reported in previous studies that have established LMAN's role in preventing deafening-induced song degradation (40–100% lesion extent, *Brainard and Doupe, 2000*). Finally, although the LMAN lesions we made largely prevented deafening-induced song degradation, birds that received LMAN lesions still showed extremely subtle but significant song degradation following deafening, as previously reported (*Figure 8A*) (*Brainard and Doupe, 2000*; *Horita et al., 2008*). Therefore, neither the complete abolition of auditory feedback by deafening nor the expression of small amounts of deafening-induced vocal plasticity in LMAN-lesioned birds are sufficient to drive structural changes to HVC dendritic spines.

## Discussion

Resolving how feedback-related information enters the AFP is a critical step to understand neural mechanisms of vocal learning. $HVC_X$ cells are attractive candidates for providing this information because they provide auditory input to the AFP (*Roy and Mooney, 2009*), receive inputs that are weakened by deafening (*Tschida and Mooney, 2012*), and may receive inputs from DAF-sensitive neurons, including HVC interneurons (*Sakata and Brainard, 2008*; *Keller and Hahnloser, 2009*). Furthermore, some forms of mammalian cortical plasticity initially involve changes to inhibitory rather than excitatory synapses (*Froemke et al., 2007*; *Keck et al., 2011*), raising the possibility that DAF might alter inhibitory input onto $HVC_X$ cells prior to any changes in excitatory input. Therefore, although the singing-related action potential output of $HVC_X$ neurons is insensitive to DAF (*Kozhevnikov and Fee, 2007*; *Prather et al., 2008*), the possibility remained that these cells integrate input from feedback-sensing cells during singing at subthreshold levels, which could then trigger slower changes to synapses on $HVC_X$ neurons as a prelude to vocal plasticity (*Tschida and Mooney, 2012*). By directly measuring synaptic activity in singing birds, we found that singing-related synaptic activity in $HVC_X$ neurons lacks real-time sensitivity to feedback perturbation. Moreover, many $HVC_X$ cells that we recorded displayed auditory responses in non-singing states, including a subset of cells that exhibited auditory-motor mirroring, a property which has been theorized to arise through integration of real-time feedback signals (*Hanuschkin et al., 2013*). Therefore, although $HVC_X$ cells receive fast synaptic inputs from several sources that may have the potential to convey auditory feedback during singing, these inputs appear to be gated to an 'off' position during vocalization in zebra finches (*Shea and Margoliash, 2003*; *Cardin and Schmidt, 2004*; *Coleman et al., 2007*).

One potential concern is that a synaptic feedback 'gate' onto $HVC_X$ cells may only open after the bird has experienced frequent and prolonged exposure to DAF. Several observations allay this concern. First, the hit rate for applying DAF we used during intracellular recordings (i.e., ~50%) has been shown here (*Figure 1—figure supplement 3*) and by others to be sufficiently frequent to drive adaptive changes in pitch (*Andalman and Fee, 2009*; *Charlesworth et al., 2012*; *Ali et al., 2013*). Second, all of the analyses of subthreshold DAF sensitivity we performed were conducted after the bird had been exposed to DAF for at least an entire preceding day, the time required for the bird to habituate to the exposure, as determined by the decreased frequency of song truncations we observed in response to DAF. Finally, because we recorded from individual birds for 1 to 2 weeks and exposed them to DAF throughout much of that time window, many of the cells we recorded from were monitored after the animal had experienced many days or even more than a week of DAF. That is, our intracellular recordings and analyses of subthreshold DAF sensitivity were made in conditions where the bird's exposure to DAF was both frequent and prolonged. And although we did not track any single cell's activity while DAF exposure induced song plasticity, a recent study by *Ali et al. (2013)* has shown that singing-related HVC multiunit activity remains unchanged even after contingent DAF is used to drive changes in song pitch. Therefore, we support a model in which auditory synapses onto $HVC_X$ cells are always gated off when the bird is singing, while also recognizing that experiments involving longer-term intracellular recordings in birds that are undergoing DAF-induced song plasticity are required to fully resolve this issue.

The absence of any detectable synaptic trace of auditory feedback in $HVC_X$ cells during singing raises the question of how deafening drives changes to synapses on these cells. One possibility is that auditory feedback information is first transmitted to $HVC_X$ cells and then to downstream error correction circuits in the AFP, but that this information transfer occurs slowly and only when the bird is not singing. Regardless of the plausibility of such an offline feedforward model, it predicts that removal of auditory feedback independent of downstream error correction processes in the AFP should be sufficient to

trigger changes to dendritic spines in HVC. In contrast, we found that LMAN lesions prevent the shrinkage and destabilization of HVC$_X$ dendritic spines that normally follow deafening. Therefore, deafening-induced changes to HVC$_X$ spines depend on an intact AFP, rather than resulting simply from the removal of auditory drive to HVC, thus raising the possibility that HVC receives error-correction signals from the AFP (*Ali et al., 2013*). Although the current experiments cannot resolve whether deafening-induced changes in HVC are driven solely by changes in LMAN activity or instead by LMAN acting in concert with the removal of auditory inputs to HVC$_X$ cells, the total insensitivity of HVC$_X$ cells to acute feedback perturbations suggests to us that deafening-induced synaptic changes in HVC$_X$ cells are a consequence rather than a cause of error correction processes mediated by the AFP. The indirect, recurrent pathways that can convey activity from LMAN to HVC (*Hamaguchi and Mooney, 2012*) provide a substrate over which the consequences of error correction processes in the AFP could be transmitted back to the early stages of the vocal motor hierarchy. Furthermore, because LMAN can access multiple stages of the motor hierarchy in tens of milliseconds (*Hamaguchi and Mooney, 2012*) and song may only change in response to altered feedback over hours or days, the consequences of performance evaluation may be widely distributed in the song motor network even before behavioral changes are first evident.

A remaining important question is how and in what form auditory feedback information enters the AFP to facilitate error correction. Previous studies have shown that input and output cells of the AFP (HVC$_X$ and LMAN neurons) are not real-time sensors of auditory feedback (*Leonardo, 2004*; *Kozhevnikov and Fee, 2007*; *Prather et al., 2008*), and the current study shows that even those feedback-dependent changes that can be detected in HVC depend on the AFP rather than fast synaptic signaling onto HVC$_X$ cells from feedback-sensing auditory neurons. These findings suggest that HVC is downstream of the performance evaluation provided by real-time feedback sensors, and also raises the possibility that these sensors communicate error-related information to the AFP independent of HVC. Notably, the ventral tegmental area provides input to Area X (*Lewis et al., 1981*) and receives indirect input from several auditory regions, including the HVC shelf, that contain DAF-sensitive cells (*Wild et al., 1993*; *Fortune and Margoliash, 1995*; *Vates et al., 1996*; *Keller and Hahnloser, 2009*). In this scenario, the consequences of error detection would reach the AFP in the form of neuromodulatory signals from the VTA, an arrangement that would buffer the entire song premotor circuitry from the role of acute feedback processing. In fact, this buffered feedback architecture, along with the gating of auditory inputs to HVC, may be necessary when the frequency of motor control is so fast that a relatively slow feedback signal becomes uninformative for the ongoing motor command.

## Materials and Methods

### Subjects

All experiments were carried out in accordance with a protocol approved by Duke University Institutional Animal Care and Use Committee. Data were collected from 14 adult (age range, 85-150 post hatch days) male zebra finches (*Taeniopygia guttata*): n = 11 for intracellular recordings in freely singing birds, n = 3 for HVC longitudinal imaging combined with LMAN lesion and deafening.

### Intracellular recordings in singing birds

Intracellular recordings were made in zebra finches using a modified version of a microdrive (*Figure 1—figure supplement 1*, original design by M Fee, MIT, *Long et al., 2010*) constructed by stereolithography (Agile Manufacturing, Inc, Ontario, Canada). A miniature motor (Part # 0206A001B+02/1 47:1-Y2825; Micromo, FL) was used to move the micropipette attached to the shuttle, and two additional stainless steel screws were used to stabilize the movement of the micropipette. A miniaturized headstage (by IY, equivalent to HS-2A headstage with gain x0.1, Axon Instruments [now Molecular Devices, CA]) was mounted on the back of the base to send signals and receive command current from the intracellular recording amplifier (AxoClamp-2B; Axon Instruments) through a flexible tether cable (Omnetics, MN). The microdrive was surgically implanted over HVC using stereotaxic coordinates. All recordings were made from the right hemisphere. HVC$_X$ neurons were identified by antidromic stimulation of Area X with a bipolar stimulating electrode (a pair of 75 µm diameter silver wires, ~500 µm apart, A-M systems, WA) implanted into Area X, or by their spontaneous, DC-evoked, and singing-related activity. Recordings were attempted for approximately 7–21 days per bird and recording pipettes were changed daily.

## Data acquisition for intracellular recordings

Custom MATLAB software with xPCtarget toolbox (by KH) (Mathworks, MA) was used for recording the microphone signal and membrane potential data, real-time detection of the target syllable, and generation of distorted auditory feedback [DAF] (*Figure 1—figure supplement 2*). In most intra-cellular recording sessions, we randomly selected a subset of songs as 'catch' trials for which DAF playbacks were suppressed (target hit rate 50%, actual hit rate, 30–70%). The microphone and membrane potential data were simultaneously recorded at 44 kHz for feature detection, down-sampled to 22 kHz, and saved for post-hoc analysis.

## Detection of target syllable and execution of DAF

For the online detection of the pre-target syllable, spectral features were calculated in short (5 ms) segments of sound and updated at 1000 Hz (*Figure 1—figure supplement 2*, step 1, 2). Before the experiments, a support-vector machine (SVM) algorithm was used to define the pre-target syllable (SVM algorithm with a Gaussian kernel using soft margins was written by KH in MATLAB). During the experiment, the same SVM algorithm and the same parameter sets (written in Simulink and xPC Target, MATLAB by KH) were used for the online detection of the pre-target syllable (*Figure 1—figure supplement 2*, step 3). Once the pre-target syllable was detected, spectral features including sound amplitude, mean frequency, Wiener entropy, FM, and pitch values were measured for a duration ~100 ms (*Figure 1—figure supplement 2*, step 4). Because the occurrence of the target syllable was almost completely predicted by the detection of the pre-target syllable in zebra finch songs, we defined the occurrence of the target syllable simply as the time when sound amplitude crossed a threshold. The occurrence of the target syllable was confirmed in post-hoc analysis. To avoid false positives (triggering DAF by detecting a natural fluctuation of syllable features), the criteria need to be met for more than 3 ms before executing DAF playback; therefore, the minimum latency between the detection of target syllable and the delivery of auditory feedback is 3–4 ms. We set the sound amplitude slightly above the level where birds tend to truncate their song (~65 dB at the center of the cage) to establish the saliency of DAF. Birds gradually begin to sing motifs of normal duration and tempo even with DAF, and our analysis is limited to the songs without truncation and delayed syllables except for *Figure 6*. Song truncation was detected by the absence of expected syllables following DAF onset. Delayed syllables were detected by referencing the expected distribution of the syllable onset timings in catch conditions with the probability p(τ | catch), where τ is the syllable onset timings relative to DAF. By randomly selecting the hit trials with the probability of p(τ | catch)/p(τ | hit), we made the syllable onset timing distributions in hit and catch conditions equivalent.

In separate experiments, we used sound amplitude and mean frequency as the contingency criteria (*Figure 1—figure supplement 2*, step 4). In this case, the bird can escape from DAF by slowly changing its syllable structure toward a certain direction of spectral features, such as higher or lower frequency. These experiments confirmed that our perturbation protocol can shift spectral features of target syllables when DAF was applied in a contingent manner as previously reported (*Tumer and Brainard, 2007*; *Andalman and Fee, 2009*) (*Figure 1—figure supplement 3*).

## Analysis of bird song for intracellular recordings

To precisely align intracellular recording data to the singing data, we first analyzed the microphone data to identify distinct syllables. The mean and variance of spectral features (sound amplitude, entropy, FM, mean frequency, pitch, goodness of pitch) during each sound event were saved in a MySQL database (http://www.mysql.com). We classified them into mutually exclusive labels, such as introductory notes, syllables, and cage noise, using an SVM-based classification algorithm (*Chang and Lin, 2001*), using a Gaussian kernel with soft margins implemented in a Matlab-based GUI interface written by KH. Manually classified data were used to train the SVM algorithm until classification accuracy exceeds 98% correctness, estimated by the cross-validation method.

## Analysis of intracellular data

To measure subthreshold DAF sensitivity, raw voltage measurements were first median-filtered to eliminate spikes (5 ms window). The filtered membrane potential data were then grouped into similar voltage ranges to ensure that average membrane potential did not differ between hit and catch conditions. For each voltage group, neurons from which we obtained at least 14 motifs of associated membrane potential data in total were used to measure DAF sensitivity. The differences of the

baseline-subtracted $V_m$ generated from catch and hit trials averaged over fast and slow time windows (20–60 ms and 60–200 ms) were compared using $t$-tests. Here the baseline-subtracted $V_m$ is defined as $V_m^i(t|S) - baseline^i$ where $V_m^i(t|S)$ is the median filtered voltage trace at time $t$ in $i$–th trial in condition S = {'hit', 'catch'}. The baseline of $i$–th trial was calculated as the mean of $V_m^i(t|S)$ during −200 to 0 ms relative to DAF onset. The difference of the mean of baseline-subtracted $V_m$ ($\Delta V_m$) are defined as follows:

$$\Delta V_m(t) = \frac{1}{M}\sum_i^M \left(V_m^i(t|hit) - baseline^i\right) - \frac{1}{N}\sum_j^N \left(V_m^j(t|catch) - baseline^j\right).$$

To calculate the mean firing rate and mean membrane potential before and during singing, mean values of the median-filtered voltage were calculated from a 0.5-s period before song (−1 to −0.5 s before bout onset) and a 0.5 s period during singing (0–0.5 s after bout onset). Here, the onset of a song is defined as the first introductory note or syllable proceeded by at least 1 s of silence. For each cell, up to the first 50 bouts that occurred during the recording were used for calculating mean firing rate and resting potential. We excluded the period from −0.5 to 0 s before song onset from these calculations because all HVC neurons recorded here displayed sharp transitions in firing rate and/or membrane potential within this period. Spikes were detected as events where the membrane potential deviates >+30 mV from the high-pass filtered membrane potential (>400Hz).

## Synaptic onset analysis

Depolarizing post-synaptic potentials (dPSPs) were detected using a modified version of the algorithm previously reported (**Ankri et al., 1994**). Briefly, the onsets of dPSPs were defined as the time points where dV/dt exceeds a threshold defined as 0.8–1.5 times higher than its standard deviation. The threshold was set relatively high; therefore, our analysis is limited to sharp-rising dPSPs. Other parameter values were set to dt = 4 ms and refractoriness = 5 ms. Because zebra finch songs are highly stereotyped, the onset timing of each dPSP is highly stereotyped across bouts, and dPSP onsets naturally form clusters in dPSP raster plot (**Figure 2C**). To identify each dPSP onset cluster, we used a $k$-means clustering algorithm to identify dPSP onsets. The number of clusters is given by inspecting the dPSP histogram and raster plots. The time range for this analysis is 20–200 ms after DAF onset. We did not include cells with overlapping dPSPs because the identification of the corresponding dPSPs across hit and catch conditions is difficult.

## Measurement of auditory responses

To quantify the auditory responses of the intracellularly recorded neurons, we first measured the subthreshold (median-filtered voltage trace relative to baseline) and suprathreshold (firing rate) activity during and before playback of the bird's own song (BOS), a stimulus known to evoke robust and selective responses in HVC neurons (stimulus duration was 1–2.4 s, played to birds at random intervals of 15-60 s). We analyzed data from neurons with 7 or more trials and the significance of each neuron's response was evaluated by either a statistical test for differences in median value (Mann–Whitney $U$-test) or variance (Ansari–Bradley test). The Ansari–Bradley test is applicable only when there is no significant difference in median value; therefore, we first tested for a significant difference in the median response using the Mann–Whitney $U$-test and then applied the Ansari–Bradley test if there was no significant difference in median response. Cells that exhibited a significant difference in either the median or variance were defined as 'responsive'. The threshold for statistical significance was set to p=0.025 for both tests. Response index of variable $r$ ($V_m$ or firing rate) is defined as follows;

$$\text{Response index} = |r_{response} - r_{response}| / |r_{response} + r_{base}|,$$

where, $r_{response}$ and $r_{base}$ are the mean or standard deviation of variable $r$ measured during response and baseline periods, respectively.

The expected playback amplitude was subtracted from the sound amplitude recorded through the microphone to detect and eliminate trials when the bird vocalized during BOS playback. The sound amplitude was set to around 60 dB at the center of the cage during day-time recordings (light on) and ~45 dB during night-time recordings (light off).

We classified our dataset based on both recording time and the bird's behavioral state. For day-time recordings, the bird's wakefulness was monitored by video, and we excluded data from periods when the bird closed its eyes and perched quietly for >1 min. For night-time recordings, the light was

turned off and we excluded the trials when the bird was making noise. The light was turned on at a fixed time each day (9AM) and was turned off around 9 PM (±2 hr; day/night ratio between 14:10 and 10:14 hr).

## Deafening and quantification of song degradation

Male zebra finches (120–150 dph) were anesthetized with isoflurane and deafened by bilateral cochlear removal. Undirected song was recorded continuously starting at least 2 days before until at least 1 week after deafening. Wiener entropy and entropy variance (EV) of each syllable in a bird's song were quantified using Sound Analysis Pro (*Tchernichovski et al., 2000*). 30 examples of each syllable were measured on each day, and values from two pre-deafening days were pooled to obtain a baseline distribution for each syllable. The onset of song degradation for each bird was defined as the day on which the distribution of values for either the entropy or EV of any syllable differed significantly from the baseline distribution and remained significantly different on all subsequent days (one-way ANOVA).

## Fluorescent labeling of HVC neurons and in vivo two-photon imaging

As previously described (*Tschida and Mooney, 2012*), GFP-lentivirus (~1 µL; eGFP expressed under the control of the Rous Sarcoma Virus LTR [FRGW]) was injected into HVC and Fast Blue or Alexa-Fluor 594 conjugated dextran amine were injected into Area X and RA (64-160 nL) in isoflurane-anesthetized male finches 4 days to 2 weeks prior to imaging. The birds were placed on a reverse day–night cycle 1 week before the first imaging session to minimize effects of imaging on their daytime behavior and were imaged longitudinally starting 1–2 nights prior to deafening. On the first night of imaging, birds were anesthetized with isoflurane and placed in a stereotaxic apparatus. A headpost was affixed to the skull using dental acrylic, and bilateral craniotomies (1–2 mm$^2$) were made over HVC. The dura was excised, and a custom-cut coverslip (No. 1 thickness) was placed over the pial surface and sealed in with dental acrylic. The birds were placed on a custom stage under a Zeiss Laser Scanning Two-Photon Microscope 510. Only GFP-labeled neurons within a field of retrogradely labeled neurons were classified as HVC neurons and imaged. Dendritic segments of identified HVC neurons were imaged twice nightly at 2 hr intervals (Zeiss Laser Scanning Two-Photon Microscope 510, 1024 × 1024 pixels, 76 × 76 µm$^2$ image size, 3.2 µs/pixel, averaging 2 samples per pixel with 1 µm z-steps, using a 40x/0.8NA Zeiss IR-Archoplan immersion objective).

## Image analysis

Three-dimensional image stacks were smoothed using a Gaussian filter (ImageJ); brightness and contrast adjustments were not made for data analysis, although images were contrast-enhanced for figure presentation. Dendritic segments to be analyzed were selected and identified in image stacks collected either 2 hr or 24 hr apart. Spine size (measured across nights, 24 hr interval) was measured as the integrated optical density of each spine head, background-subtracted (using the optical density of neuropil next to the dendrite) and normalized to the mean brightness of the adjacent dendritic shaft. Spine size index was calculated as the ratio of spine size values measured during imaging sessions separated by 24 hr (time 24 size/time 0 size). Spine stability was calculated as the percentage of spines maintained each night (2 hr interval). Data from LMAN intact, deafened birds and LMAN intact, hearing birds (previously described in *Tschida and Mooney, 2012*) are reported as raw values (i.e., post-deafening measurements of spine size index and spine stability were not normalized to baseline, pre-deafening measurements for data presentation).

## LMAN lesions

The boundaries of LMAN were mapped out using extracellular recordings, and LMAN was lesioned bilaterally by 7% ibotenic acid injection at least 4 days prior to deafening or imaging. Nissl-stained tissue from 2 LMAN intact control birds was used to estimate the average volume of LMAN, and the volume of LMAN remaining in LMAN lesion birds was divided by this value to estimate lesion extent. Additionally, injections of retrograde tracer were made into RA 4–5 days prior to sacrificing the animals to retrogradely label any remaining RA-projecting neurons in LMAN and to aid in the identification of remaining, non-lesioned portions of LMAN in Nissl-stained tissue sections.

## Source code

MATLAB and xPCtarget based data acquisition program for realtime detection of target syllable and playback of sound. This program also contains cell search mode and antidromic cell type identification

mode used to conduct intracellular recording in freely behaving birds. All the codes are uploaded in GitHub (https://github.com/hamaguchikosuke/RealSong_control).

## Acknowledgements

The authors thank David Schneider for reading earlier versions of the manuscript and Jennifer Baltzegar and Michael Booze for technical assistance.

## Additional information

### Funding

| Funder | Grant reference number | Author |
| --- | --- | --- |
| National Institutes of Health | R01 DC02524 | Richard Mooney |
| National Institutes of Health | R21 NS079929 | Richard Mooney |
| National Institutes of Health | R01 GM78031 | Bruce R Donald |
| National Science Foundation Integrative Organismal Systems | 0821914 | Richard Mooney |

The funders had no role in study design, data collection and interpretation, or the decision to submit the work for publication.

### Author contributions

KH, KAT, Conception and design, Acquisition of data, Analysis and interpretation of data, Drafting or revising the article; IY, BRD, Contributed unpublished essential data or reagents; RM, Conception and design, Drafting or revising the article

### Ethics

Animal experimentation: All experiments were carried out in accordance with a protocol (A292-11-11) approved by Duke University Institutional Animal Care and Use Committee.

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
