## [Decision Letter]

[Editors’ note: although it is not typical of the review process at *eLife*, in this case the editors decided to include the reviews in their entirety for the authors’ consideration as they prepared their revised submission.]

Thank you for sending your work entitled “Mechanisms underlying a synaptic trace of auditory feedback in a sensorimotor nucleus important to learned birdsong” for consideration at *eLife*. Your article has been favorably evaluated by a Senior editor and 3 reviewers, one of whom is a member of our Board of Reviewing Editors.

The following individuals responsible for the peer review of your submission have agreed to reveal their identity: Ronald L Calabrese, Reviewing editor; Marc Schmidt, peer reviewer.

The Reviewing editor and the other reviewers discussed their comments before we reached this decision, and the Reviewing editor has assembled the following comments to help you prepare a revised submission.

In this study, Hamaguchi et al. perform a careful electrophysiological analysis of subthreshold activity in HCV_X_ neurons using sharp microelectrode recordings of the awake unrestrained songbird to assess the role of auditory feedback during singing in these neurons. Positioned as they are in the auditory/singing pathway these HCV_X_ neuron seem a likely target for feedback during singing that leads to behavioral plasticity, but extracellular recordings have not shown an effect of acoustic feedback on spiking during singing: thus the motivation in this study to look for subthreshold effects of feedback. The results show convincingly that synaptic input to HVC_X_ cells can be driven by sound input but not during singing corroborating the previous extracellular results. These experiments are supplemented by an ablation (AFP) experiment that shows that deafening induced plasticity of spines on HVC_X_ neurons requires an anterior forebrain pathway. This structural study provides novel conclusions. This elegant analysis in freely singing birds will generate wide interest in the songbird and sensory motor integration communities.

All the reviewers thought the work was elegant and important but there were some minor concerns that should be addressed. The reviews are highly congruent and complementary, and they are provided in their entirety so that the authors can benefit from them directly. The authors should address reviewer concerns and implement all the minor edits. They should particularly address this minor concern:

“A remote possibility with DAF is that auditory responses in HVC_X_ might only occur after a window of time. One could envisage, for example, that HVC only processes auditory feedback errors if these occur for a critical amount of time. In other words, auditory flow might be gated for small infrequent errors (this might prevent the motor system from changing to any random perturbation of auditory feedback during singing) but might become “un-gated” once the perturbations are sensed as occurring in a consistent manner (this “consistency” could potentially be monitored in the auditory forebrain and need not rely on the AFP or VTA). A prediction of this idea is that HVC_X_ neurons would not show any changes in activity if (1) DAF is too infrequent or (2) when DAF is first introduced. It would be helpful for the authors to address this issue. This argument could easily be resolved if the authors are able to show that DAF-induced changes in song also occur in the birds they record from. Ideally, the most convincing argument would be to show a lack of change in HVC_X_ neurons during DAF even while the birds undergo changes in song.”

Because such a demonstration would constitute a new study they should squarely address this issue in Discussion.

Another point that the authors should address is making the title more descriptive of what the manuscript actually presents.

*Reviewer**#1*:

In this manuscript the authors perform a careful electrophysiological analysis of subthreshold activity in HCV_X_ neurons using sharp electrode recordings of the awake unrestrained songbird during singing to assess the role of auditory feedback during singing in these neurons. Positioned as they are in the auditory/singing pathway these HCV_X_ neuron seem a likely target for feedback during singing that leads to behavioral plasticity, but extracellular recordings have not shown acute feedback during singing: thus the motivation in this study to look for subthreshold feedback. The results show convincingly that synaptic input to HVC_X_ cells can be driven by sound input but not during singing corroborating the previous extracellular results. These experiments are supplemented by an ablation experiment that shows that deafening induced plasticity of spines on HVC_X_ neurons requires an anterior forebrain pathway. This structural study provides novel conclusions. This elegant analysis in freely singing birds will generate wide interest in the songbird and sensory motor integration communities.

Concerns:

The writing is succinct but the paper is not always easy to follow and could benefit from more explanation for the general reader. The work is carefully done with detailed analysis and the figures appropriately illustrate the data. Nevertheless the figures are very complicated and could benefit from more detailed legends and perhaps better labeling. A better general figure of the system than Figure 1 would help, including some of the detail of Figure 7 up front. Essential information from the supplemental figures should be incorporated into the text according to *eLife* policy – I defer to the expert reviewers on what is essential.

*Reviewer**#2*:

The authors convincingly demonstrate their two main points: lack of subthreshold responses to auditory feedback perturbations in HVC_X_ neurons and a dependence of deafening-induced changes in spine morphology on an intact AFP. The work represents a very high level of technical proficiency and, except in rare cases, the presentation is very clear.

My primary concern centers on the impact of the paper’s findings. As the authors correctly point out, a number of prior studies have used extracellular recordings to show that the pattern of action potentials fired by HVC_X_ neurons is insensitive to perturbations of auditory feedback. The first main finding of the present manuscript *–* that subthreshold activity is also not affected by feedback perturbations *–* therefore confirms current thinking in the field and appears to have limited interest except to specialists in the system.

The paper’s second main finding – that post-deafening changes in spine morphology in X-projecting HVC neurons depends on an intact AFP – has somewhat higher impact. Earlier work from the same lab (Tschida and Mooney, Neuron 2012) convincingly demonstrated that deafening-induced changes in spine morphology precede the degradation of song and that the changes in spine size predicted the magnitude of behavioral changes. These earlier results raise the question of whether the morphological changes reflect the sensory perturbation (deafening) or the beginning of AFP-dependent motor changes that are first detectable days later. The present results demonstrating that spine changes depend on an intact AFP show that such changes are not solely a function of altered auditory feedback, ruling out a simple model (Figure 7) in which changes in spine morphology depend only on the removal of ascending auditory inputs. Although ruling out this model leaves several important questions unanswered (especially the mechanisms by which patterns of AFP activity could regulate spine changes in HVC and how LMAN’s indirect inputs to HVC might interact with ascending auditory signals), it nevertheless represents an advance in knowledge over prior studies.

In summary, I have no substantial concerns about how the research was performed or the validity of the results. My primary concern is that the first finding described above does not substantially change current thinking about the system. The second finding advances our understanding of sensorimotor plasticity by ruling out a simple model of auditory dependence and therefore will be of greater interest to readers.

*Reviewer**#3*:

This is a very carefully executed experimental study investigating, in songbirds, how and where auditory feedback modifies vocal motor circuits. This is a question that lies at the core of how auditory feedback shapes vocal learning. Over the past several decades, a number of increasingly refined experiments have shown that auditory feedback is necessary for vocal learning and that perturbations of auditory feedback will cause, even the normally stereotyped song of adult zebra finches, to degrade. The time course over which this degradation occurs is not immediate. Depending on the experimental paradigm and the precision with which song is analyzed, changes in song acoustic structure can be observed within hours to days. Because lesions of a specialized basal ganglia circuit (known as the anterior forebrain pathway (AFP)) prevents song degradation following feedback perturbation (e.g., deafening by cochlear removal or distorted auditory feedback (DAF)), it has been hypothesized and shown that song degradation is an active process that can be prevented by interrupting this basal ganglia circuit, for example with lesions of its output nucleus, LMAN.

One of the primary assumptions has always been that auditory feedback information is processed by nucleus HVC. This nucleus receives direct input from the auditory forebrain, is necessary for song production and was thought to provide the only known auditory input to the basal ganglia circuit by way of neurons that project directly to the basal ganglia, known as HVC_X_-projection neurons. It had been assumed that these neurons would relay auditory feedback information to the basal ganglia. It was therefore surprising that these neurons did not shown any changes in firing pattern during DAF even though feedback distortion was sufficient to perturb song.

In the present study, the authors used intracellular electrodes (technically heroic experiments) to record from HVC_X_-projecting neurons during singing under both normal and DAF conditions. They provide compelling evidence that DAF does not alter (at all!) the highly stereotyped membrane potential fluctuations that are recorded during song production. This despite recent findings, by the same lab, that DAF modifies synaptic inputs onto these cells. While these results do not provide a direct answer as to where auditory feedback is first monitored in the vocal motor system, these experiments are extremely important in ruling out HVC as the site for such monitoring.

In this study the authors also show that changes in synaptic strength (as measured by spine dynamics) of HVC_X_-projection neurons following auditory feedback perturbation (here they use deafening) are dependent on LMAN. They use this finding to argue that auditory feedback dependent changes in HVC_X_-projecting neurons are dependent on LMAN and that HVC is therefore downstream of where the real-time sensors of auditory feedback. Because the authors have previously show that LMAN receives direct auditory input from the VTA, they suggest that this circuit might be a possible candidate for auditory feedback processing, but they do not show any evidence for this.

While the authors do not, in the end, show where auditory feedback processing occurs, this study is extremely important because it makes a strong case that it does not occur in HVC. The paper is well written and the experiments are performed with great care and the conclusions sounds.

Concerns:

A remote possibility with DAF is that auditory responses in HVC_X_ might only occur after a window of time. One could envisage, for example, that HVC only processes auditory feedback errors if these occur for a critical amount of time. In other words, auditory flow might be gated for small infrequent errors (this might prevent the motor system from changing to any random perturbation of auditory feedback during singing) but might become “un-gated” once the perturbations are sensed as occurring in a consistent manner (this “consistency” could potentially be monitored in the auditory forebrain and need not rely on the AFP or VTA). A prediction of this idea is that HVC_X_ neurons would not show any changes in activity if (1) DAF is too infrequent or (2) when DAF is first introduced. It would be helpful for the authors to address this issue. This argument could easily be resolved if the authors are able to show that DAF-induced changes in song also occur in the birds they record from. Ideally, the most convincing argument would be to show a lack of change in HVC_X_ neurons during DAF even while the birds undergo changes in song.

---

## [Author Response]

We thank the three reviewers for their helpful comments, and we are grateful for their constructive criticism. We have done our best to follow their various suggestions, as detailed below. As instructed by the Reviewing editor, we have addressed Reviewer 3’s concern about “ungating” in the Discussion. We have also changed the title to render it more descriptive of what the manuscript actually presents.

Reviewer #1:

*The writing is succinct but the paper is not always easy to follow and could benefit from more explanation for the general reader*.

We have done our best to edit the manuscript for clarity and accessibility, in part by providing more explanation for the general reader in the Introduction, Methods, Discussion, and figure legends.

*The work is carefully done with detailed analysis and the figures appropriately illustrate the data. Nevertheless the figures are very complicated and could benefit from more detailed legends and perhaps better labeling. A better general figure of the system than*Figure 1*would help, including some of the detail of*Figure 7*up front*.

We modified Figure 1 for clarity by including some of the details from Figure 7. We also included more complete labeling of song system structures. We have tried where possible to provide more detailed legends, and to further simplify or expand the figures.

*Essential information from the supplemental figures should be incorporated into the text according to eLife policy – I defer to the expert reviewers on what is essential*.

We have incorporated information from the supplemental figures into the text to conform to *eLife* policy.

Reviewer #2:

*My primary concern centers on the impact of the paper’s findings. As the authors correctly point out, a number of prior studies have used extracellular recordings to show that the pattern of action potentials fired by HVC*_*X*_* neurons is insensitive to perturbations of auditory feedback. The first main finding of the present manuscript – that subthreshold activity is also not affected by feedback perturbations – therefore confirms current thinking in the field and appears to have limited interest except to specialists in the system*.

*The paper’s second main finding – that post-deafening changes in spine morphology in X-projecting HVC neurons depends on an intact AFP – has somewhat higher impact. Earlier work from the same lab (Tschida and Mooney, Neuron 2012) convincingly demonstrated that deafening-induced changes in spine morphology precede the degradation of song and that the changes in spine size predicted the magnitude of behavioral changes. These earlier results raise the question of whether the morphological changes reflect the sensory perturbation (deafening) or the beginning of AFP-dependent motor changes that are first detectable days later. The present results demonstrating that spine changes depend on an intact AFP show that such changes are not solely a function of altered auditory feedback, ruling out a simple model (*Figure 7*) in which changes in spine morphology depend only on the removal of ascending auditory inputs. Although ruling out this model leaves several important questions unanswered (especially the mechanisms by which patterns of AFP activity could regulate spine changes in HVC and how LMAN’s indirect inputs to HVC might interact with ascending auditory signals), it nevertheless represents an advance in knowledge over prior studies*.

*In summary, I have no substantial concerns about how the research was performed or the validity of the results. My primary concern is that the first finding described above does not substantially change current thinking about the system. The second finding advances our understanding of sensorimotor plasticity by ruling out a simple model of auditory dependence and therefore will be of greater interest to readers*.

We respect the reviewer’s perspective on the novelty and/or potential impact of the intracellular data we collected in singing birds that fails to detect evidence of auditory feedback perturbation. However, we believe that the clear negative results we obtained with these intracellular recordings are critical to distinguishing between two different models of auditory feedback processing in songbirds. Together, the intracellular recordings and imaging datasets we obtained allow us to advance a model where the song premotor elements in HVC are buffered from feedback during singing. This model stands in contrast to that developed by Sakata and Brainard (albeit in a different species) and also clarifies pathways by which deafening affects synapses on song premotor neurons (45). As the third reviewer noted, “…these experiments are extremely important in ruling out HVC as the site for such [feedback] monitoring.”

Reviewer #3:

*A remote possibility with DAF is that auditory responses in HVC*_*X*_* might only occur after a window of time. One could envisage, for example, that HVC only processes auditory feedback errors if these occur for a critical amount of time. In other words, auditory flow might be gated for small infrequent errors (this might prevent the motor system from changing to any random perturbation of auditory feedback during singing) but might become “un-gated” once the perturbations are sensed as occurring in a consistent manner (this “consistency” could potentially be monitored in the auditory forebrain and need not rely on the AFP or VTA). A prediction of this idea is that HVC*_*X*_* neurons would not show any changes in activity if (1) DAF is too infrequent or (2) when DAF is first introduced. It would be helpful for the authors to address this issue. This argument could easily be resolved if the authors are able to show that DAF-induced changes in song also occur in the birds they record from. Ideally, the most convincing argument would be to show a lack of change in HVC*_*X*_* neurons during DAF even while the birds undergo changes in song*.

The reviewer brings up an interesting idea, namely that a synaptic feedback “gate” onto HVC_X_ cells may only open after the bird has experienced frequent and prolonged exposure to DAF. Several observations allay this concern. First, the hit rate for applying DAF we used during intracellular recordings (i.e., ∼50%) has been shown here (Figure 1—figure supplement 3) and by others to be sufficiently frequent to drive adaptive changes in pitch (2; 8; 1). Second, all of the analyses of subthreshold DAF sensitivity we performed were conducted after the bird had been exposed to DAF for at least an entire preceding day, the time required for the bird to habituate to the exposure, as determined by the decreased frequency of song truncations we observed in response to DAF. Finally, because we recorded from individual birds for one to two weeks and exposed them to DAF throughout much of that time window, many of the cells we recorded from were monitored after the animal had experienced many days or even more than a week of DAF. That is, our intracellular recordings and analyses of subthreshold DAF sensitivity were made in conditions where the bird’s exposure to DAF was both frequent and prolonged. And although we did not track any single cell’s activity while DAF exposure induced song plasticity, a recent study by [1] has shown that singing-related HVC multiunit activity remains unchanged even after contingent DAF is used to drive changes in song pitch. Therefore, we support a model in which auditory synapses onto HVC_X_ cells are always gated off when the bird is singing, while also recognizing that experiments involving longer-term intracellular recordings in birds that are undergoing DAF-induced song plasticity are required to fully resolve this issue. These potential concerns and the points made above that address them are now considered in the second paragraph of the Discussion, as directed by the Reviewing editor.